# Cruelty against Leniency: The Case of Imperial Zoroastrian Criminal Law

Janos Jany

Faculty of Humanities and Social Sciences, Pázmány Péter Catholic University, 1088 Budapest, Hungary; jany.janos@btk.ppke.hu

**Abstract:** The article examines the impact of Zoroastrianism on criminal law and legal theory during the reign of the Sasanian dynasty (224–651 C.E) in late Antique Persia. This was the historic period when Zoroastrianism was also the ideology of the Iranian state, which granted the Zoroastrian church extraordinary power and influence, a unique situation which is termed by the author as 'imperial Zoroastrianism'. The first part of the paper examines how imperial Zoroastrianism evolved from previous understanding of religion and law. The second part of the paper scrutinises the Zoroastrian understanding of wrong in the light of eschatology and cosmology, and the ethical principles that follow from this very particular world view. Next, an individual section is devoted to the criminal theory of Zoroastrianism, which regards criminal punishment not as a punishment but as a means to save the soul of the offender from sufferings in Hell. With such an underlying principle in mind, the text looks for examples of cruelty and leniency in substantive criminal law and criminal procedure. This main body of the article examines contemporary legal sources and apocalyptic works. Finally, a comparison of Hindu and Islamic criminal legal theory follows the description of the Zoroastrian criminal law, highlighting astonishing similarities. Considering the results of both the analytic and the comparative methods, the author comes to the conclusion that it is not religion in itself that suppresses crimes, but rather their eschatology and cosmology: religions that are based on divine justice are less lenient toward crimes and offenders than religions in which alternative concepts like divine grace or non-violence are also operative.

**Keywords:** Zoroastrianism; Sasanian Persia; criminal theory; criminal law





## 1. Introduction: What Is Imperial Zoroastrianism

Zoroastrianism is one of the oldest religions in the world, its history going back to three millennia, at least. Like other old religions, Zoroastrianism has thus many faces due to the changes it observed during its long history. It is next to impossible to cover all historic periods of Zoroastrianism in merely a short article. For this reason, here I will only focus on the Sasanian period (3rd–7th centuries AD); which was in many aspects the most dramatic and successful era for the followers of prophet Zarat.ushtra.[1]

Throughout these long centuries, a Zoroastrian church was formed and developed by outstanding priests like Tansar and Karder (3rd century), a hierarchical organisation that had never been seen before in Iran. This hierarchically formed body of local, provincial and chief mowbeds (priests) was designed according to the model of contemporary Iranian structure of public administration, corresponding to local, provincial and central (royal) administration. It was certainly not the only cooperation between the Iranian state and the Zoroastrian Church. Since the founder of the Sasanian dynasty, Ardashēr (226–240) gained power by rebellion and successful military campaigns against the lawful ruler of the country, the last king of the Parthain Arsacid dynasty had great need to legalise his rule.[2] Zoroastrianism or, properly speaking, Mazdaism (Mazda yasna = Mazda-worshipping: from the name of chief god Ahura Mazda), came to his rescue, and formed an alliance with the king that is discernible both in literary texts (political and religious) and artistic

representations found on rock reliefs and coinages. Royal glory (khwarenah), one of the basic political concepts of Ancient Iran, is represented by a round shaped object symbolising the Sun. It is seen in Ardeshēr's inauguration rock relief, and was to be handed over by Ahura Mazda himself. Coins, too, bear witness to the strong cooperation between state and religion: there are religious symbols on all coins (mainly fire altars with priests).[3] This symbiosis, both theoretic and practical, resulted in the so-called twin theory, according to which the state and religion are twins born from the same womb and never to be separated, a credo that was most likely formulated some centuries later but projected back to the third century.[4]

Needless to say, there was a good price to be paid for such a religious backing for a dynasty that started its career with a coup d'etat. The price was a fair share in political power and the judicial administration to which the monopoly of intellectual professions and religious propaganda should be added.[5] There is a discrepancy between what religious texts claim (as part of their own propaganda) and what contemporary political reality appears to be. For example, according to texts of Sasanian political theory (composed by Zoroastrian priests) the heir to the throne was elected from the legitimate sons of the ruling king, and it was the nobility and the high priest who took part in the selection, but in any case of a dispute it was the high priest who had the last say. In reality, however, no Sasanian king ever tolerated such interference in his own family business and appointed his successor by himself. In doing so, he certainly consulted the nobility and the priesthood, but consultation is a far cry from decision-making. It is also true, however, that should he choose a bad heir, that is, someone whose politics was against the interest of the noble families and the Zoroastrian church, such a ruler was soon deposed or killed.[6] On the other hand, in judging judicial cases brought to the king as the final instance, in both criminal and private legal matters the monarch consulted the chief mowbed, his legal advisor ex officio. We can assume that he followed his counsel, though no one could force him to do so.[7]

Despite its huge influence in politics, administration and intellectual life, Zoroastrianism was never declared to be the official state religion of the Sasanian realm as one might expect it to be, given the twin theory, or as a reaction against the law of Theodosius, which obliged everyone in Rome to embrace Christianity (380 AD). Consequently, other religions remained tolerated, though less so than in the previous centuries of the Achaemenids and the Arsacids, where dynasties granted nearly unlimited religious freedom to non-Zoroastrians (Dandamaev and Lukonin 1989, pp. 347–60). Sasanian religious policy was influenced by both the mowbeds and their religious zealotry coupled with domestic and foreign policy (M. Stausberg 2001, pp. 235–44), a tendency that spared the Babylonian Jewish community from suffering atrocities (J. Neusner 1966) but led to the prosecution of Iranian Christians, who from time to time were falsely accused to be on the side of the Roman enemy (J. P. Asmussen 1983, pp. 924–48). In short, Zoroastrianism was not an officially declared state religion but a highly favoured ideology of the Sasanian dynasty, a curious and unique situation in the history of Zoroastrianism for which I now use the term imperial Zoroastrianism. This period is very important for researchers because the majority of our sources, legal and religious, were produced during these centuries. For believers, however, the legacy is ambivalent, because though glorious and prosperous, it was simultaneously corrupted by power and wealth, making it subject to criticism and disdain from contemporary believers.

Imperial Zoroastrianism contributes to the administration of crimes, punishments and the mitigation of sanctions in four different, though interrelated, aspects. First, as a religion, it defines ethical standards and condemns unethical behaviour as every religion does. Since highly immoral deeds are simultaneously crimes (murder, theft, etc.), the mutual relationship between religion and law is obvious in such situations. Second, as a religion in power, it also defines particular offences that are also classified as crimes by law, which are mainly against its power position but have nothing to do with religious ethics as such. Apostasy, for example, that is, leaving the Zoroastrian community for the sake

of another religion, is not a wrong in absolute terms, only through the lens of a religious administration that would want to avoid a decrease in the number of believers even by force, if necessary (the same law was applied, most probably following the Zoroastrian example, by Muslims). Another example is the "offence" of cohabiting with a non-Iranian (Zoroastrian) woman; an act which is null and void in terms of family law but also amounts to a crime.[8] Third, ritual-centred Zoroastrianism created a great number of condemned acts (some of which also amounted to crimes) which are neutral from the perspective of power and ethics but are highly important for a ritualistic world view. A woman walking in the rain, or a woman sitting in water[9] are completely neutral deeds except for in a ritualistic approach which is obsessed with ritual purity and the horror that a menstruating woman might defile a sacred element such as water. Fourth and, probably, the most important aspect is that Zoroastrian priests took part in the administration of the judiciary, a complex system of which the details are not entirely clear to us due to the absence of relevant sources. Despite this, it is quite clear that there were both royal and ecclesiastic courts in the Sasanian empire, the latter controlled by the priesthood, which had competencies in legal cases related to Zoroastrianism (M. Macuch 1981, pp. 188–208). Such cases were mainly criminal law matters (apostasy, the persecution of Christians) coupled with some legal cases of inheritance that were important for the nobility and the priesthood (e.g., chagar marraige, stūrīh) to preserve the wealth in their families (M. Macuch 1995, pp. 149–67). In this way, Zoroastrian law ceased to exist only as a legal system of a religious community; instead, it was transformed to an imperial level, an unprecedented development in Iranian history that made non-Zoroastrian persons also subject to Zoroastrian law, at least in some criminal law matters. Here, the twin theory comes close to social reality because state and religion acted in concert: judgments of ecclesiastic courts could be brought to the king, as the final instance in all legal matters, who decided in person (at least, in important cases) but never without the help and strong involvement of the chief mowbed, while final judgments were always executed by state authorities (prisons, the military, hangman, etc). This situation very much resembles the Spanish inquisition of the late 15th century, the aim of which was to enforce Christian belief in a newly created state by an institution that was the strong arm of the Spanish court and responsible for it (this is why it operated independently of papal jurisdiction).

## 2. The Zoroastrian Understanding of Wrong

Zoroastrianism is above all an ethical religion, the teaching of which concentrates on decisions humans have to make between good and evil throughout their entire lives. Zoroastrian teaching does not concede individual discretion to decide what is wrong and what is not but defines it in every possible detail. Cosmogony and old Indo-Iranian concepts such as arta (asha) help Zoroastrian thinkers articulate their understanding of good and bad, as these are not subjective categories but are anchored in nature and the divine world.[10]

According to the very long tradition of cosmogony, the material world was created by chief god Ahura Mazda (Lord Wisdom) and his agents (the seven Amesha Spentas) who were responsible for separate acts of creation (e.g., the creation of plants, animals, humans, etc.). Their world was, however, corrupted heavily by Angra Mainyu (Evil Spirit), chief opponent to the supreme deity, and also called the Lord of Lies (draugha).[11] Here we have the personification of good and bad placed into a larger framework of cosmogonic speculation, which makes it possible to classify every human deed into simple binary categories. Every act that supports the divine aspect of creation is considered good, while everything opposed to it is regarded as bad. In addition to human acts, objects and living beings are also classified into these two categories. For example, animals that are created by divine forces are considered good, such as dogs and cattle, while cats, scorpions, mice, snakes, etc., are seen as evil creatures (khrafstra), and must be treated accordingly. In fact, they are killed and a festive occasion is dedicated to this, together with a particular

instrument made for this purpose (khrafstra-gan), specified in a work called Law against the demons (Vidēvdād).[12] Souls of persons who killed many reptiles enjoy glory in Heaven.[13]

This evident dualism places a heavy burden on humans who find themselves in the centre of a cosmic struggle in which they participate creatively because every human act strengthens either the forces of God or those of Evil. Needless to say, as creatures of God, humans have to take side with Ahura Mazda and must not commit any evil deeds. Things are, however, more complicated because of the assault the Evil Spirit launched on the creatures of the divine. As a result, descendants of the primordial man, Gayomard, are corrupt creatures who are free to choose to commit sin, which they in fact did from the very first generation onwards. As a result, a wrongful deed is not only a breach of ethical standards or widely accepted social norms, but an act in favour of Evil in its struggle against Truth (arta) and the divine and natural order. Since every deed is classified as either good or bad, all human failures are interpreted as strengthening the Evil Spirit and its cohorts. In such an ongoing cosmic struggle, the free will every human enjoys is of utmost importance (Zaehner 1976, pp. 67–79); a prerequisite to divine justice every soul has to face after death. Zoroastrian symbolism leaves no doubt about the centrality of the good behaviour in its religious understanding. In addition to fire altars and the figurative representations of Ahura Mazda, the "three words" are written on every religious building, including modern Zoroastrian temples as far away as the United States.[14] The three words are "humat, hūkht, huwarsht"; that is, good thoughts, good words and good deeds; encapsulating the main ethical message of Zoroastrianism.

Antagonists of good order are considered to be agents of the Lord of Lies (draugha) who act on the behalf of the Evil Spirit. It is important to note that humans are free to choose this path but will be treated accordingly when facing divine judgment by gods Rashnu and Mithra, who complete their jobs impartially regardless of social status, political function or wealth. The soul of a dead man departs from the Earth at the dawn of the fourth day accompanied by divine helpers (*Srosh*, *Vahram*) and demons of different kinds (e.g., the demons of death and wrath). Divine judgment could sentence a soul either to Heaven or to Hell depending on good or bad thoughts, words and deeds. After divine judgement, the soul has to go through a bridge where it will meet a young, never before seen beautiful maiden or a hideous and ill-favoured wench; depending on its fate, it goes either to Heaven or Hell. The beautiful maiden is the personification of the good deeds of the deceased while the ugly creature is the manifestation of its own bad deeds.[15] The fate of the soul in Hell is full of suffering; punishments in Hell are brutal, described in some texts with almost sadistic pleasure, most vividly in the *Arda Virāz Nāmag*, a Pahlavi text originating from the late Sasanian or the early Islamic period.

Hell is not a place of eternal torment but a place for temporal punishments for all the evils one has committed on Earth. Such an understanding derives from both the inherent logic of the religion, emphasising divine justice, and the teaching of the final judgment. The notion of divine justice only allows for temporal sufferings because of the principle of proportionality, a prerequisite of justice. As the evil deeds of a person could be numbered, the punishments for these deeds should also not be eternal. The second reason for limited sufferings in Hell is final judgment at the end of the historic time when the material world as we know it will cease to exist. This is the end of history, when the world will be cleaned of all evil and corruption, caused by the Evil Spirit and his allies, and will be purified. This is a key concept in Zoroastrian thinking, called making wonderful (Frashgird). The rising of Soshyans, a descendant of Zarat.ushtra, marks the beginning of this period accompanied by catastrophic events. More importantly, this is the time when all the dead will rise and be judged. This was conceived as the model of an ordeal going back to centuries: humans, and their resurrected bodies united with their souls, must go through the molten metal ordeal for three long days. This ordeal inflicts no harm on the righteous but causes hardship for the wicked. This is also the time of the final battle between Ahura Mazda and his adversary and it is the supreme deity that emerges as the winner. The only point that has divided Zoroastrian theologians is the fate of the wicked: while some thought they would perish

because of their evil deeds, others believed they would be purified, join the righteous and praising God (M. Boyce 1975, pp. 240–46; Sh. Shaked 1994, pp. 36–39; Zaehner 1976, pp. 139–44).

Though a place of only temporal suffering, Hell is nevertheless a place of horror, as described in the *Arda Virāz Nāmag*, a book devoted to visions of Heaven and Hell. Accordingly, Virāz, a righteous man, was allowed a spiritual journey in which he gained insight into the state of affairs of both Heaven and Hell. While the description of Heaven is rather brief, the sufferings in Hell are portrayed in detail, covering the majority of the book's content. In Hell, Virāz sees souls suffering from various torments; these are described very naturalistically and each torment is for a certain type of sin. In this way, the author also identifies the most grievous sins for which torments are inevitable. What is more, torments in Hell resemble the sin or the crime committed; thus torments also incorporate symbolic values which are attached to certain crimes but not to others. Some examples will clarify matters: (1) the punishment of a woman who was a whore and practiced sorcery was this: her tongue would be cut out, her eyes gouged out, snakes, scorpions, worms and other reptiles (all Satanic creatures!) would eat her brain and from time to time she would chew her own flesh; (2) the punishment for a woman that insults her husband with her sharp tongue was this: her tongue would be branded; (3) the punishment of a man who copulated with a woman during her menstruation period was this: his mouth would be constantly filled with the filth and the menstruation of women and he would continuously cook and eat his own child; (4) the punishment of a woman who aborted her child and destroyed it and left it as a carrion would be this: she would have to dig into a mountain with her breast while she had a stone on her head in the shape of a corpse.[16] Needless to say, it is in the interest of all human beings to escape such sufferings, even when one has committed grievous sins for which Hell seems to be the proper punishment. It is exactly at this point where criminal law enters the scene.

### 3. The Zoroastrian Understanding of Crime and Punishment

It seems, admittedly, a little bit bizarre that criminal law in general and physical punishments in particular could rescue a wicked person; but it is in fact the case in Sasanian Zoroastrianism. As demonstrated above, the original idea was divine justice according to which every human soul is sentenced to what it deserves, without any chance of mitigating the consequences. Later, however, this notion was modified and some techniques were introduced in order to reduce the torments of the soul in Hell. Founding pious organisations to finance important ritual acts (fire worship) and offerings and prayers for the soul were such devices (J. Jany 2004, pp. 269–86), others being close kin marriage (khwēdōdāh) (M. Macuch 1991, pp. 141–54), a controversial yet very commonly propagated union within the family of believers; and criminal law was the third technique to help souls out of their miserable situation.

It is thus clear now that the object and aim of criminal law was neither retaliation against crime nor the protection of society from the offender, but to help souls avoid horrible punishments in the afterlife. Penitence for sins and punishments for unlawful deeds evolved as part of a central understanding of criminal law because according to the underlying religious principle, penitence and punishment not only mitigate the sufferings in Hell but annul the offence (sin) for which the penitence and the punishment was administered. In other words, if a wrongdoer was punished for a bad deed in the material world, the very same deed will not be taken into consideration in divine judgment. What is more, the beneficence of his previous good deeds can be felt again and not be overshadowed by evil deeds any longer.[17]

Confession should be announced truthfully and openly, as we learn from the *Shāyast nē-Shāyast*, an early post-Sasanian work about a variety of ritual and legal topics; to confess openly means that the offender has to enumerate all his sins without omitting any of them, while to confess truthfully means that he should declare that he would not sin again.[18] Publicity, however, is also an important element of the confession, because the



delinquent has to confess in the presence of the victim (or his/her relatives) if an offence was committed against a person, or in the presence of a priest if the offence was a violation of a religious norm,[19] a dichotomy to which we shall come back later. After confession, all the good thoughts, words and deeds of the offender could be taken into account in the divine tribunal. Should they, however, not confess their misdeeds, they could be outlawed and if they die in this period, they are regarded as *margarzan*, that is, persons who commit the most grievous crime.[20]

The last threat should be taken seriously because *margarzan* sinners who do not confess their crimes are sentenced to Hell, regardless of their good thoughts, words and deeds, a position clearly at variance with divine justice. Even after the final judgment, their head would be cut off for every crime they committed and subsequently brought to life again in order to inflict extra suffering, punishments and torments.[21] When reading this paragraph, it becomes clear that it is not the crime itself that results in endless suffering, but the lack of confession and punishment. Sadly, the fate of those who confessed their crime openly was not any better because, as we will see soon, their punishments were sometimes as frightening as those in Hell.

Obviously, a verbal confession was only the first step to avoid divine punishments in the afterlife. This is why criminal punishments in the mundane world were executed with those who confessed their offences. At this point too, the twin theory was realised without hesitation, as criminal procedures in such cases were administrated by competent priestly tribunals and their sentences were executed by state authorities, which reflects a tacit approval of their judgments. Presiding priest-judges, called rad, had unlimited power and no secular power could limit their competences. This is because they were more than priests administrating criminal cases in the name of a powerful church and state; they were spiritual authorities, a function that goes back millennia, that is, to Avestan times or even earlier.[22] Competencies of priests and judges could be limited but that of a spiritual leader could not; otherwise, it would harm the very function of the spiritual authority. Here, the old, Indo-Iranian tradition concerning the role of a spiritual authority, the twin theory and its realisation in the judiciary, and the theology of criminal law came together in a synergic way that reinforced each component. Middle Persian texts kept this no secret when ruling that *margarzan* sinners should hand over all their wealth and body to the rad[23]; which meant a confiscation of their wealth and also their family, the members of which were considered as part of the family wealth.[24] To hand over the body was a blank cheque for the rad to pronounce any sentence he saw fit for the benefit of the soul at the divine tribunal. Should it be a capital punishment, his sentence was executed, in the majority of the cases, by beheading with the sword.[25]

To complicate matters, offenders were considered ritually impure, an additional difficulty in a ritual-centred religion. Ritual impurity was a great obstacle for social and profession life and, therefore, individuals were interested in overcoming ritual impurity rapidly and becoming clean again. There were abundant rituals to cure this social hindrance; performing the required rituals together while reciting the accompanying prayers was enough to become ritually pure again. However, for offenders, the situation was complicated because it was their deed itself that made them impure and the only way out was confession and penitence. This is what Zoroastrian scholars unanimously claimed as the only point of debate on the fate of *margarzan* sinners. According to the majority view, *margarzan* sinners too, become ritually pure after their confession, penitence and subsequent punishment, but there were dissenters who denied this by saying that margazan sinners always remain impure,[26] a rigorous view at variance with divine justice. We have to bear in mind, however, that all these things came into play only when an offender went to a rad and confessed his crime. The horrific scenes in Hell and the continuous ritual impurity of an offender without confession were the only means of bringing criminals to court in a state in which any kind of a police force was missing.

What were, then, these grievous *margarzan* crimes for which one had to lose wealth, family and life in order to being released from its consequences? To answer this question,

we have to immerse ourselves in the Sasanian typology of crimes. Zoroastrian scholars classified crimes in the framework of two different systems. One such a system was a simple dichotomy between offences called wināh i ruwānīg (against the soul) and wināh i hamēmālān (against the adversary). It is our misfortune that the texts now available do not tell us to which categories individual offenses belong, a state of affairs that hinders our understanding, since the categorisation seems to be at odds with the logic of imperial Zoroastrianism: if every wrong affects the fate of the soul, then a category "against the soul" has no meaning because misdeeds against adversaries also influence the outcome of the divine judgment. Perhaps, as Maria Macuch argues (M. Macuch 2003, pp. 173–74), this system came into being long before Sasanian times and the terminology remained intact at times when it was a bit confusing. In the absence of a clear categorisation in the sources, we may assume that offences against adversaries are deeds that non-religious systems would define as crimes, that is, acts harming other persons' lives (murder), health (any kind of injuries), social position (slander) or wealth (robbery), while misdeeds against the soul are acts that harm basic principles and laws of Zoroastrianism, mainly in the field of ritual purity and justice (e.g., corrupt judges), and family life (a disobedient wife). If the sadistic punishments in Hell observed by Virāz were to be meted out to persons who committed offenses against the soul, then we can construct a list of such misdeeds. Studying all the offences specified in the *Virāz* Nāmag, there is in fact not a single offence that could be understood as aggression against fellows, such as murder, any kind of physical injury, robbery, rape, etc.

Here are the offences (some of them mentioned multiple times) that are punished in Hell:

(a) Telling lies (text P. 29, 31, 53);
(b) Sorcery (text P. 29, 48);
(c) Heresy (text P. 30, 33, 42);
(d) Pollution and negligence in use of water and fire (text P: 30, 37);
(e) Cheating of partners and workers, the poor and travellers, breaking contracts (text P. 31, 34, 35, 41, 47);
(f) Theft (text P. 33);
(g) Going frequently to the bath (text P. 31);
(h) Abortion and not nursing a child (text P. 32, 38, 40, 47, 52);
(i) False testimony (text P. 37);
(j) Killing pious (mardōm i ahlav) people (text P. 37);
(k) A woman lamenting and weeping, having a sharp tongue, quarrelling with her husband, beautifying herself and refusing intercourse (text P 38, 39);
(l) Adultery by both sexes (text P. 38, 39, 40, 43, 48–49);
(m) Sodomy (text P. 44);
(n) Hurting parents (text P. 41);
(o) Killing and tormenting cattle (text P. 45–46);
(p) Corrupt judge (text P. 47, 50–51);
(q) Disrespect for rulers and the army (text P. 54).

As might be expected from a strongly patriarchal society, as Sasanian no doubt was, sins committed by women are abundant, ranging from adorning their body to distorting family life and to killing children. It is, however, also worth noting that adultery of husbands was also not tolerated, at least, in divine judgment.

We may ask, what is the common denominator between these offences, which at first glance seem very diverse? There is a sharp logic in this enumeration when we realise that all offences are basically about the neglect of the social cohesion of a group, be it a natural one (family), a religious one (Zoroastrian community), or a political one (the state). Subjects of such offences are the basic rules of (a.) religion itself (ritual purity laws, heresy, killing pious men); (b.) family life (disobedient wife, a careless mother); (c.) political obedience (disrespect for rulers and army commanders); and, perhaps, worse of all, (d.) justice (a corrupt judge, false testimony, telling lies, cheating of partners). All these could be interpreted within only a particular group with a degree of social cohesion

and the offences break this very social cohesion and confidence. That is, a treacherous business partner ruins the honor of his partners and workers, an adulterous woman ruins the confidence of her husband, while a corrupt judge ruins the confidence in his offence. If *Arda Virāz Nāmag* predominantly collected the offences called wināh i ruwānīg, then these are in fact against the soul, not only against that of the offender but also against the "soul" of any social group, that is, cohesion and confidence.

The second categorisation of crimes is a more complicated one, comprising eight different categories (framān, āgrift, ōyrisht, ardush, khwar, bāzāy, yāt, tanāpuhl). There are some important notes to be made here. First, it should be emphasised that these are not individual crimes but categories of offences to which individual crimes are assigned. Second, the listing also reflects a hierarchy, starting with the less important misdeeds and ending with the most severe category (tanāpuhl). Third, there is "mobility" between the categories, but only from the inferior to the superior and never the other way around. The reason for such an upward movement is either time or frequency. For example, if a crime belongs to the category yāt, it is considered as such for the first time. Should, however, the same offence be committed three times or over a long period of time (one year), the offence would be assigned to a higher category, that is, to the next in the hierarchy, tanāpuhl. It is also important to note that such mobility was observable only between neighbouring categories, that is, from the original position to the next category but never to two or more categories. Forth, a clear categorisation is also missing here: we do not know for sure what crimes belonged to each of these categories. Some of them are known because of the casuistic references to them, for example, to cohabit with a non-Iranian woman is khwar[27] while the laziness of a disciple who does not learn the sacred texts by heart is ardush.[28] The most documented category is tanāpuhl, consisting of a variety of ritual offences (a bleeding woman walking in the rain, walking without the sacred girdle, to talk while eating, etc.) and also apostasy.[29]

Fifth, the most horrible crimes by offenders, called margazan (worth death), are missing from both frameworks and there is no explanation for this astonishing fact in the sources and we are also helpless to find an answer. It does not mean that *margarzan* were unimportant; *margarzan* were the offences of persons who committed crimes that had to be punished by death, without any sign of leniency. Some of these deeds were offences that were originally only tanāpuhl but after the passage of time (one year), they became more serious. One such a tanāpuhl offence was apostasy, which amounted to *margarzan* after one year without the intent to return back to the Zoroastrian community. This logic of the typology of crimes is in perfect accordance with historic sources, claiming that apostates were sentenced to death after a year in prison where they had enough time to think about their future and to listen to Zoroastrian priests who "educated" them, a cynical reference to inquisition (Iqbāl 1942, p. 22).

Persons declared to be *margarzan* injured the two most important objects of imperial Zoroastrianism: the authority of the fire and that of the king. Zoroastrians are mistakenly called fire-worshippers (ātashperestān) as it is not the fire as a physical object that they praise, but instead, the fire is only a symbol of Truth (arta), the underlying principle of the divine order. This is why an attack against the fire, that is, ritually improper behaviour, is at the same time a defilement of the principle itself. There is no need to extensively explain why an offence against the person of the king and the political order of the monarchy in general brought retaliation via the strongest means of criminal law, a legal situation to be found in every state. It is also a matter of fact that in a monarchy based on the twin theory an attempt against a king is at the same time the most grievous sin in terms of religion. Seen from this angle, it is understandable why persons who fled the battlefield or attempted to take the life of a king were *margarzan* and treated accordingly.[30] *Margarzan* were also those who carried a corpse (ritually the most unclean object) toward the fire so that the corpse was enflamed, who carried a corpse in the rain or a corpse alone or threw a corpse into water, or a pregnant woman who ate carrion. By contrast, it seems unbalanced to declare

someone *margarzan* because of killing an animal without proper recitation of the sacred text (Nērang) or for failing to recite the *Gatha*s because of drunkenness.[31]

To this list we can add politically motivated criminal procedures against dissenters, mainly Christians and Manicheans. As we have seen above, apostasy was a tanāpuhl crime which amounted to *margarzan* after a year, ending with decapitation, but the majority of the Christians executed were not apostates, but Christians by birth, which was far from a crime. The penalisation of religious rivalry was a new policy in Iran, the government of which had offered religious freedom to Jews and Christians for long centuries. Even Mani, the founding prophet of Manicheism, enjoyed the hospitality of the second Sasanian king, Shābuhr I, to whom his main work was dedicated (Shābuhragān). Though Asmussen emphasises the religious zealotry of mowbeds in the prosecution of Christians (Asmussen 1983, pp. 924–48), it is also worth noting that hard times for Christians began only after Christianity became the state religion of the main adversary, Rome; this too was when diplomatic relations deteriorated and the outbreak of war between the rival empires was impending. Since these trials were administered by ecclesiastic courts (rarely presided over by the king in person), the power position of the Zoroastrian clergy did not result in softening criminal procedures and punishments but rather, led to the opposite: criminal punishment was seen as a strong arm of the religion which cemented the political system based on the twin theory, even by unlawful means if necessary. Priestly extremism was manifested, for example, in the case of Pūsai, who was sentenced to death; but it was the king who made every effort to rescue his very talented court administrator from the wrath of the mowbeds because he needed his skills and professionalism (Braun 1915, p. 70).

A brief look at criminal procedural law also reinforces our impression that it was cruelty and not leniency that determined the operation of the criminal machinery. Though here we can find an astonishingly modern idea, bail, which is an established institution of criminal procedural law in the common law countries, yet very unprecedented in ancient Near-Eastern legal cultures. We do not know how common it was to temporarily release accused persons from gaol on bail as we have only few cases as proof, but what is clear from the trial of the Christian martyr Narseh is that some persons (perhaps only prominent ones) were released from prison between the sessions of the court with the obligation to return to the next hearing. To guarantee his return, the accused had to sign a contract with an official in charge. In addition, a bailsman was also appointed with the obligation to guarantee the personal appearance of the accused persons. These details rather affirm the assumption that bail was an established legal institution and Narseh's case was not an ad hoc decision (G. Hoffmann 1880, pp. 37–38; Braun 1915, pp. 145–46).

However, we cannot find institutional policies that favoured the accused other than bail. In fact, the whole procedure was rather repressive, even painful and deadly. Though accused persons were given the right to defend themselves at the court hearings, not everyone was still alive at this point of the proceeding since those accused were severely tortured during investigation. This is because confession was seen as the most prestigious evidence—similarly to the pre-modern European thinking—and investigators tried hard to "convince" the accused to confess. Contemporary prison wardens did not restrict their fantasies of how to torment accused persons who were put in gaol between the court hearings when the trial was composed of more than one hearing. We can find heavy beatings among the most frequent torments, several times a day, placing the accused among robbers and murders, denial of food, drink and clothes and visitation of relatives. Now, it is clear what a favour it was to be released on bail or to be put on house arrest. When a capital punishment was pronounced, it was executed without much delay, all the more so if it was the king who made the final decision. In addition to decapitation and crucifixion, burning alive, cutting into pieces, killing with arrows and stoning were the most frequent methods of execution.[32]

It would be unfair, however, to make Zoroastrianism responsible for all these cruelties. It was not the case that the Zoroastrian–Persian criminal praxis stood out as an exceptionally cruel one among more humanistic and lenient neighbouring systems. By contrast, neither

Roman nor Islamic criminal machinery proved to be less sadistic, not to mention the shocking cruelties of the former neo-Assyrian realm and its terror, which provoked hatred in the entire ancient Near-East. In the same vein, pre-modern European criminal procedure was also not impregnated with humanistic thoughts; otherwise the mission of Voltaire and Beccaria, among others, would have been meaningless. In short, religious influence was minimal in criminal procedural law and the criminal machinery was determined instead by contemporary usage and practices.

## 4. Comparing Neighbouring Systems

Before concluding it would be wise to look around and see how neighbouring legal systems, with which Zoroastrianism has a shared history, think about crime and punishment. Hinduism is obviously our first subject because of its common ancestry with Zoroastrianism.

Due to its later and separated development, the internal structure of Hinduism differs enormously from that of Zoroastrianism (e.g., the concept of karma, which is alien to Zoroastrianism, a cyclic understanding of time against the linear time frame of Zoroastrianism, a ritualised social hierarchy, the varna system, which is lacking in Iran, etc.), yet even so, there are some similar notions. Among these similarities is the centrality of pollution and purity and interestingly, the almost identical understanding of crime and punishment. Ritual purity also has a central role in Hinduism, but its operation is complicated by the varna system. This complexity is very much discernible in criminal theory, which is not to be expressed in general terms, only on a case-by-case basis. For example, it is impossible to say how much a Brahmin is affected by a crime because it depends on him being an agent or a victim. If a Brahmin is an offender, his pollution is minimal, while if he is a victim, the pollution of the offender is at the maximum. As a general principle, it can be said that the wider the social distance between the victim and the offender, the more severe the pollution of the offender, together with his punishment. Obviously, why members of the highest varnas should be more privileged than others has nothing to do with "pollution" itself but reflects social and political dominance.[33] Despite this complexity, the basic notion that offences (crimes and sins) result in ritual pollution is a common understanding to be found in both Hinduism and Zoroastrianism.

The next common feature is the cooperation and sharing of power between secular and religious authorities following the line of the Persian twin theory. This is why an offence can be both a crime and a sin and why it could be met by punishments and expiations. Legal sanctions are administered by the secular authority, that is, the king and the royal courts; expiations are administered by the religious authorities, the Brahmins and their tribunals. To further stress institutional similarities, it is important to note that kings were advised in legal matters by their legal advisors, the chief priests (purohita, chief mowbed), who were better versed in law, and rulers were therefore advised to follow their instructions. Despite this theoretic dependency, royal courts remained secular courts in both India and Persia with kings presiding over them.[34]

The most striking parallel is, perhaps, the function of punishment as a salvation for the soul of the offender. Manu made no mistake in emphasising this understanding: *But men who have done evil and have been given punishment by kings become free of defilement and go to heaven, just like people who have done good deeds.* (Book VIII: 318) (Doniger and Smith 1991, p. 186). No Zoroastrian mowbed could summarise his own vision of a criminal punishment more elegantly. As a result, it is a sacred duty of kings to punish offenders according to law, this being part of their own dharma (rajadharma).[35] To do otherwise would be adharma for them, at the same time threatening the fate of the soul of the offender. Needless to say, the varna system also made this basic principle more complicated, because according to Brahmanic ideology, a crime of a Brahmin that is left unpunished causes him less suffering in the afterlife compared to a sudra whose crime also remains unnoticed (Glucklich 1986, p. 107). However, this is nothing more than an attempt to enlarge social dominance to the salvation theory of punishment, which left the basic idea untouched.

By contrast, Zoroastrianism is more "democratic" in treating humans equally in this regard, instead concentrating on the very act of the offence itself and not on the person of the offender. As a result, priests were not exempt from the consequences of the unpunished wrong, yet the system also remained blind toward persons who would deserve some positive discrimination. These people did not escape the attention of Hinduism, however, its emphasis being on the human agents who commit sins (crimes). As a result, particular attention has to be given to women, the sick, the poor, infants, aged persons and persons of diminished intellect while determining their punishments.[36] This sign of some kind of a leniency is an astonishing gesture toward the underprivileged that cannot be deduced from Brahmanic ideology, which is obsessed with social standing and pollution.

Such considerations are completely alien to Islam, a less ritualistic religion in which criminal punishment is not a means to salvation but rather a mundane consequence of an offence. The situation is more complicated, however, because Islamic criminal law is separated into three different layers of norms, each with its own logic and principles. To make a very complicated and long story short: there are offences that are committed against the law of God (haqq Allāh) called hudūd, such as adultery, theft, highway robbery, etc.; offences that are mainly injuries committed against fellows, called jināyāt, which go back to pre-Islamic tribal customs of Arabia; and ta'zīr, crimes defined by political authorities (Caliphs, governments), offences that are absent from both hudūd and jināyāt. What is important here is the difference between hudūd and jināyat both in their logic and principles.

Jināyāt is a category for a variety of offences that are committed against fellows and can be punished only upon the request of the victim and his/her relatives if the victim has passed away (valī al-dam). A Western lawyer would call such processes accusatorial because they start only upon the accusation of the victim and what organs of the judiciary do is to state guilt or innocence and act accordingly. Consequently, the master of the process is in fact the victim (or his relatives) who can stop the proceeding any time by pardoning the offender and favouring a compromise instead. What is more, courts even prefer such a friendly conclusion and ask victims about their intention in different phases of the proceeding. The final judgment of the courts is, interestingly, not the final word in the process, as after pronouncing a capital punishment, the court asks the victim for the last time whether he wants to pardon the offender (afw) against the payment of blood money (diya) or if he insists on the execution, exercising his right to retaliation (qisās) (al-Misrī 1994, p. 587).

Proceedings in relation to hudūd offences, by contrast, are not accusatorial because the intention of the victim has no place whatsoever in the process. This is because these offences are believed to be committed against the law and authority of God itself, and the fact that there are "mundane victims", too, plays little role. This is why there is little room for leniency by the authorities and forgiveness by the victim of the offence. For example, if one steals an object from anyone, the victim is not able to stop the criminal procedure by pardoning the offender. If the victim happens to forgive the thief when learning his miserable standards of living, he cannot stop the legal proceeding by declaring his pardon because the criminal procedure follows its own logic after it has been set in motion. If guilty, the thief would lose his hand, even against the will of his victim, because the logic of the procedure is inquisitorial and not accusatorial.[37] On the other hand, if a highway robber confessed his offence before the criminal procedure started, he would be no longer subject to hudūd, but nevertheless have to pay compensation for the loss of lives and wealth of the victims (al-Misrī 1994, p. 616). In other words, confession transforms the offence from hudūd to jināyāt and with this, to a completely different logic of criminal law.

Now time has come to remember the Zoroastrian dichotomy between wināh i ruwānīg and wināh i hamēmālān, translated (interpreted) as offences against the soul and against fellows, a categorisation that fits perfectly to the Islamic model. I do not claim, of course, that the Islamic system was developed according to the Zoroastrian model, only that there is a striking parallel between the two bodies of offences: one committed against

religious precepts and the other against fellows. Unsurprisingly, their logic is also the same: proceedings for offences against adversaries are accusatorial and subject to pardon and leniency at any time; procedures in relation to offences against religious laws are strict, inquisitorial and not subject to moderation or pardon. Humans can pardon offenders but God cannot or, properly speaking, institutions acting in the name of God cannot (see, for example, apostasy, which results in capital punishment in both imperial Zoroastrianism and Islam). With this, time has come to conclude.

## 5. Conclusions

As outlined above, imperial Zoroastrianism was certainly not a religion that preferred leniency in criminal procedure. It should be emphasised, however, that this was not because of a particular cruelty or a stubborn execution of laws, but the result of the eschatology of Zoroastrianism.

Zoroastrianism is a strongly ethical religion that adds ritual purity and pollution as important elements to its worldview. The struggle between good and evil can be observed in every aspect of religious thinking, from cosmology to eschatology, and of the central position of mankind in the cosmic struggle between forces of God and those of the Evil Spirit, the Lord of Lies. Humans have to choose in all their deeds, between the truth and the false and all their deeds reinforce either the power of God or his adversary. In such a constant struggle, wrong deeds or what is more, corrupt persons, cannot be tolerated since they support the forces of Evil which threaten heavenly harmony in its essence. In the Zoroastrian thinking, it is not the person but rather the deed itself that is in the focus: it does not condemn the offender but most of all the offence; that is, the wrong in the ethical and the pollution in the ritual understanding cannot be pardoned, otherwise one would succour the force of Evil, an unthinkable idea. A religion which interprets its own mission as central in the cosmic struggle against the forces of Evil simply cannot tolerate illegal and unethical deeds.

In addition to the ethical consideration, the role of punishment in eschatology is the next hindrance to leniency and forgiveness in criminal procedure. As we have seen, the execution of the proper and just punishment was interpreted as a means to escape from the more brutal torments in Hell. Seen in this light, punishment is a licence for offenders who would otherwise suffer divine punishment, the consequences of which being more brutal than those of mundane criminal procedure. Sadistic scenes in Hell described in the *Arda Virāz Nāmag* explained how judgments in the world to come had to be understood without any hint of leniency and mercy. In such an eschatology, the absence of a criminal punishment (and atonement) or its improper administration (that is, a more strict or more mild punishment) would endanger the afterlife of the offender and therefore, be against his existential interest, at least in the long run. Zoroastrianism stops here and does not stretch the problem further to the liability of institutions (priests, kings, royal courts) that fail to do justice for whatever reasons (corruption not included because corrupt judges have their punishment in Hell). Hinduism, by contrast, brings this logic to its extreme when claiming that a king who does not punish an offender according to law, but instead pardons him, threatens not only the salvation of the offender but also his own, because in such a situation the king does not follow his own dharma (rajadharma) but acts against it (adharma).

The obsession with ritual purity and pollution is the third factor that hinders leniency toward criminal offenders, though this is less serious a hindrance than eschatology and cosmology. Thus, it is only an additional circumstance anchored instead in social hierarchy and dominance rather than in religious dogmas. In short, ritual pollution and all its consequences are not real reasons against leniency, only reinforcing the ethical and eschatological arguments.

Finally, there were political reasons too, in times of imperial Zoroastrianism, a consideration which is only due for the Sasanian period (224–651) and not for the entire history of Zoroastrianism. Threatened by both internal (Manicheism) and external (Christianity) rivals, Zoroastrian authorities did everything to protect their imperial status and to reduce

the influence of religions that were seen as potential challengers for influence and power. Politics as a factor became more influential after the conversion of Rome to Christianity, which was hence seen as the ideology of the main enemy. Unsurprisingly, in criminal procedures against Iranian Christians, mercy and leniency were not options. Politics continued to have its impact on Zoroastrian criminal law, though the other way around; after the Muslim conquest, Zoroastrians were no longer able to execute their judgments as they were not privileged with the ius gladii and, obviously, no Muslim authority executed the judgements of Zoroastrian courts. As a result, offences were not abolished but remained intact in legal theory, but not in legal practice; needless to say, Muslims were unwilling and Zoroastrian was unable to execute a former Zoroastrian who converted to Islam because of committing apostasy.

In sum, imperial Zoroastrianism was not a religion advocating leniency toward offenders. In addition, neighbouring religions Hinduism and Islam share the same attitude. Can we then claim that religions in general are against the idea of leniency and pardon in criminal procedural law? Certainly not, if we recall the history of Burmese criminal law in which capital punishment was already abolished in the 18th century by the Manugye Dhammathat (1762), one of the most important codes in Burmese history. Here, the influence of Buddhism is beyond any shade of doubt, since the text itself refers to Buddhist ethics, highlighting the principle of non-violence. Accordingly, a bad and unlawful deed (murder) cannot be made worse with another killing (execution of the offender) because such a decision would enhance the number of evil acts on Earth. Kings who do not condemn anyone to death, even offenders, are, therefore, worthy of praise by both men and gods, whose reward is happiness and prosperity. As humans pardon themselves, they also have to pardon others.[38]

Now it is clear that it is not religion in itself that is against leniency and pardon, but rather the cosmology and eschatology of various religions. In religions that are based on divine justice, there is in fact little room for leniency, which would endanger the operation of the entire system. Justice requires the proper verdict for everyone, be it good or bad. It is the very act itself which is in the centre of such thinking, because these are against the law (will) of a higher entity. By contrast, in religions that in addition to divine justice also emphasise other principles such as divine grace and mercy (Christianity) or non-violence and pardon (Buddhism), the religious obstacle against a more human criminal law is reduced considerably.

**Funding:** This research received no external funding.

**Institutional Review Board Statement:** Not applicable.

**Informed Consent Statement:** Not applicable.

**Data Availability Statement:** Not applicable.

**Conflicts of Interest:** The author declares no conflict of interest.

## Notes

[1] This article is based on my previous research on Zoroastrian criminal justice, the result of which was published in *Iranica Antiqua* (Jany 2007, pp. 347–87). The author is grateful for the suggestions and remarks of the unknown reviewers and also for the invitation of the editor to contribute to this volume.

[2] For more on Sasanian history, see the recent works of Josef Wiesehöfer (2001, pp. 153–83); Touraj Daryaee (2013).

[3] See the works of authorities like Robert Göbl and Ryka Gyselen; a good survey is that of Nikolaus Schindel (2016).

[4] This is what the *Letter of Tansar*, a lost Middle Persian work claims. The text has reached us in New Persian as an insertion into Ibn Isfandiyar's Tārikh-i Tabaristān: A. Iqbāl (1942, p. 17), see also: M. Boyce (1968).

[5] For an overview of Sasanian Zoroastrianism see M. Boyce (1979, pp. 101–44); and: Boyce (1992); R. C. Zaehner (1961, pp. 175–92); most recently: Michael Stausberg (2001, Band 1, pp. 206–35).

[6] The most famous case being that of Hormizd IV (579–590) who was blinded and later killed by aristocratic conspirators.

[7] Here, the *Book of the Deeds of Ardashēr* (Kār-Nāmag-i Ardashēr) comes to mind in which there is a story that the king sentenced his pregnant wife to capital punishment, clearly a judgment against the law which prohibited the execution of pregnant women

until delivery because the child is innocent. The chief mowbed did not dare to contradict openly but did not let the woman be executed, presenting a false report to the king and rescuing the life of the child (who was destined to be the second Sasanian king). See H. S. Nyberg (1964, pp. 1–3).

8　Hērbedestān 12. 29. Text edition: Kotwal and Kreyenbroek ([1992] 2003).

9　Shāyast nē-Shāyast: 3.28. Text editions: J. Tavadia (1930); K. Mazdapur (1990).

10　Schlerath and Skjærvø (2012); see also: 'Truth' in: Almut Hintze (2007, pp. 53–57).

11　Zoroastrian cosmogony is given in detail in the Bundahishn, an encyclopaedic work written in Middle Persian, originating, most probably from late Sasanian times. The process of the creation of the material world is told with excerpts from Pahlavi sources (e.g., Bundahishn, Shkand Gumānīg Vizār) excellently by R. C. Zaehner (1976, pp. 29–80).

12　Stausberg (2001, p. 139); for details on Zoroastrian law on animals in general see Macuch (2003, pp. 167–90).

13　The vision of Heaven in F. Vahman (1986, p. 199).

14　For the migration of Zoroastrians to the new world, see J. R. Hinnels (1988, pp. 19–49); J. R. Hinnels (2005, pp. 425–542).

15　Stausberg (2001, pp. 144–50); some Pahlavi texts in English translation can be found in Zaehner (1976, pp. 134–137).

16　The vision of Hell in F. Vahman (1986, pp. 199–219).

17　Shāyast nē-Shāyast 8. 5–6.

18　Shāyast nē-Shāyast 8. 8–9.

19　Shāyast nē-Shāyast 8. 1.

20　Nērangestān, Fragard 2, 23.7 in Kotwal and Kreyenbroek ([1992] 2003, pp. 32–33).

21　Shāyast nē-Shāyast 8. 7.

22　Ph. G. Kreyenbroek (1984, pp. 1–15).

23　Shāyast nē-Shāyast 8. 5.

24　MHD 97. 15: M. Macuch (1993).

25　Shāyast nē-Shāyast 8. 6.

26　Shāyast nē-Shāyast 8. 5–6; 2. 107.

27　Hērbedestān 12. 29.

28　Hērbedestān 17. 1.

29　The casuistic enumeration of tanāpuhl offences are to be found in the second chapter of the Shāyast nē-Shāyast.

30　Kār Nāmag- i Ardashēr in: Nyberg (1964); Braun (1915, p. 43); Arda Virāz Nāmag in Vahman (1986, p. 218).

31　Shāyast nē-Shāyast 2. 9, 40, 63, 76, 81–82, 85, 91, 105; Nērangestān: Fragard I: 11. 2; II, 36.3; Hērbedestān 20.5.

32　For more on hearings and executions, see my *Criminal Justice in Sasanian Persia* (380–82) with abundant references to primary sources.

33　For more on this subject, see A. Glucklich (1986, pp. 96–102).

34　Vasistha Dharmasūtra, 16.1–3. In: P. Olivelle (1999, p. 290).

35　Āpastamba Dharmasūtra, 28.13. In: P. Olivelle (1999, p. 72).

36　Manu Book VIII. 126; Kautilya: Arthashastra: 4. 10, 17–18 in: R. P. Kangle (1969); Glucklich (1986, p. 113).

37　This is the view of the majority of the legal schools, though Hanafīs disagree: see al-Māwardī (1996, p. 247); Ibn Rushd (1996, pp. 542–46).

38　Manugye Dhammathat: Book 5, 1§. In: D. Richardson (1847).

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
