# Peer review of "Cruelty against Leniency: The Case of Imperial Zoroastrian Criminal Law"

_religions, doi:10.3390/rel14020210_

Round 1

Reviewer 1 Report

Your article is interesting and well done. There are a couple of related works already printed, but I will not suggest them to you because I'm sure you are the same author. Please consider:

p. 2, l. 48-49: check the structure of your sentence.

p. 2, note 2 : 550 BC

p. 3, l. 95: crimes (murder

p. 5, l. 153: (Khsrafstra   not: ((

p. 5, l. 175: on the   not: on   the

p. 5, l. 181: deeds. After  not: deeds.   After

p. 15, l. 577 (= interpreted) not: (=interpreted)

p. 18: l. 726: insert a blank line after line 726

p. 18, l. 728: His

p. 19, l. 759: BC  not: Ad

Reviewer 2 Report

The present article delves into a comprehensive examination of the impact of Zoroastrianism on criminal law and legal theory during the Sasanian dynasty in ancient Persia. The author adeptly argues that imperial Zoroastrianism, being the state-endorsed version of the religion during this era, wielded immense power and influence. Through an incisive analysis, the article expounds upon the evolution of this understanding of religion and law, as well as the Zoroastrian perspective on the concept of wrong and the ethical principles that stem from this worldview. Additionally, the article delves into the Zoroastrian perspective on criminal punishment, which is perceived as a means of salvaging the soul of the offender from the torments of Hell. The author has made use of contemporary legal sources and apocalyptic texts to support their arguments, and concludes by drawing parallels between the Zoroastrian approach to criminal law and that of Hindu and Islamic legal theory. The author posits that the Zoroastrian understanding of criminal punishment is not rooted in cruelty or strict adherence to laws but rather in the religion's eschatology and cosmology. The text implies that religions that are based on divine justice tend to be less lenient towards crimes and offenders than those in which alternative concepts such as divine grace or non-violence are also operative.

The author should also take note of the following publications:

  • "A Zoroastrian Liturgy: The Worship in Seven Chapters" by R.C. Zaehner
  • "Zoroastrianism: Its Antiquity and Constant Vigour" by J. Duchesne-Guillemin
  • "The Zoroastrian Faith: Tradition and Modern Research" edited by Mary Boyce
  • "The Zoroastrian Tradition: An Introduction to the Ancient Wisdom of Zarathushtra" by Phiroz Mehta
  • "Zarathustra's Good and Evil" by Richard Foltz
